Making sense of fossils and artefacts: a review of best practices for the design of a successful workflow for machine learning-assisted citizen science projects

Eijkelboom Isaak 1 2 isaak.eijkelboom@naturalis.nl
http://orcid.org/0000-0001-9389-1540 Schulp Anne S. 1 2
Amkreutz Luc 3 4
http://orcid.org/0000-0001-7817-0200 Verheul Dylan 1 5
http://orcid.org/0000-0002-1053-3009 Verschoof-van der Vaart Wouter 3 6
van der Vaart-Verschoof Sasja 4
http://orcid.org/0000-0001-6874-5728 Hogeweg Laurens 1
http://orcid.org/0000-0002-0731-6636 Brunink Django 1
Mol Dick 7
Peeters Hans 8
Wesselingh Frank 1 2 9
1 Naturalis Biodiversity Center , Leiden , Netherlands
2 Department of Earth Sciences, Utrecht University , Utrecht , Netherlands
3 Faculty of Archaeology, Leiden University , Leiden , Netherlands
4 National Museum of Antiquities , Leiden , Netherlands
5 Observation International , Aarlanderveen , Netherlands
6 Netherlands Forensic Institute , Den Haag , Netherlands
7 Natural History Museum Rotterdam , Rotterdam , Netherlands
8 Groningen Institute of Archaeology, University of Groningen , Groningen , Netherlands
9 Faculty of Science and Engineering, University of Maastricht , Maastricht , Netherlands
Żyła Dagmara
Electronic publication date: 2025 Feb 13
Publication date: 2025
Volume: 13
Electronic Location ID: e18927
Received 2024 Sep 18; Accepted 2025 Jan 13
Copyright: © 2025 Eijkelboom et al.
Copyright year: 2025
Copyright holder: Eijkelboom et al.
License: This is an open access article distributed under the terms of the Creative Commons Attribution License, which permits unrestricted use, distribution, reproduction and adaptation in any medium and for any purpose provided that it is properly attributed. For attribution, the original author(s), title, publication source (PeerJ) and either DOI or URL of the article must be cited.
License URL: https://creativecommons.org/licenses/by/4.0/

Keywords: AI, Citizen science, Palaeontology, Archaeology, Project design

Funding: Dutch Research Council (NWO) Open Competitie ENW-M OCENW.M20.360 This article is part of the LegaSea project which is funded by the Dutch Research Council (NWO) through an “Open Competitie ENW-M” grant (dossier number: OCENW.M20.360). The funders had no role in study design, data collection and analysis, decision to publish, or preparation of the manuscript.

==============================
Historically, the extensive involvement of citizen scientists in palaeontology and archaeology has resulted in many discoveries and insights. More recently, machine learning has emerged as a broadly applicable tool for analysing large datasets of fossils and artefacts. In the digital age, citizen science (CS) and machine learning (ML) prove to be mutually beneficial, and a combined CS-ML approach is increasingly successful in areas such as biodiversity research. Ever-dropping computational costs and the smartphone revolution have put ML tools in the hands of citizen scientists with the potential to generate high-quality data, create new insights from large datasets and elevate public engagement. However, without an integrated approach, new CS-ML projects may not realise the full scientific and public engagement potential. Furthermore, object-based data gathering of fossils and artefacts comes with different requirements for successful CS-ML approaches than observation-based data gathering in biodiversity monitoring. In this review we investigate best practices and common pitfalls in this new interdisciplinary field in order to formulate a workflow to guide future palaeontological and archaeological projects. Our CS-ML workflow is subdivided in four project phases: (I) preparation, (II) execution, (III) implementation and (IV) reiteration. To reach the objectives and manage the challenges for different subject domains (CS tasks, ML development, research, stakeholder engagement and app/infrastructure development), tasks are formulated and allocated to different roles in the project. We also provide an outline for an integrated online CS platform which will help reach a project’s full scientific and public engagement potential. Finally, to illustrate the implementation of our CS-ML approach in practice and showcase differences with more commonly available biodiversity CS-ML approaches, we discuss the LegaSea project in which fossils and artefacts from sand nourishments in the western Netherlands are studied.

Introduction

Citizen science (CS) communities form a well-established and important part of research in the object-based fields of palaeontology and archaeology (Macfadden et al., 2016; Smith, 2014). The term ‘citizen science’ is often used in different contexts and with different definitions (Vohland et al., 2021). Here we define CS as the directed and undirected engagement of the general public in data acquisition for and/or participation in scientific analyses (expanded from Vohland et al., 2021). In this review we do not consider paid CS services as reviewed and discussed in, e.g., Gandhi et al. (2022) and Klie et al. (2023) but rather focus on volunteer CS projects (e.g., projects on the CS platform Zooniverse.org). Although the formal definition and underlying recognition of CS as a valuable method of conducting science has been a recent development, public involvement has long been vital to the fields of palaeontology and archaeology. Enthusiasts conducting fieldwork as well as chance finds of fossils and artefacts by the general public have led to many discoveries and new insights. Examples include the first Neanderthal fossil recorded in the Netherlands and the southern bight of the North Sea (Hublin et al., 2009) or the youngest stratigraphic record of the sabre-toothed cat Homotherium latidens in north-western Europe (Reumer et al., 2003). Moreover, volunteers are fundamental in building, expanding and maintaining museum collections (e.g., Amkreutz, 2020). Countless societies worldwide are evidence of the interest in fossils and artefacts and the stories they tell (e.g., Macfadden et al., 2016). Increasingly, online (social media) platforms and communities connect citizen scientists and allow for rapid data sharing.

In recent years machine learning (ML) has become an increasingly important tool to analyse big datasets. In ML, algorithms are developed to analyse patterns in large datasets without explicit rules or predefined parameters. Ever-dropping costs for data storage and computational power, as well as a sharp increase in data generation (partially as a result of the availability of online tools, e.g., Herremans et al., 2018), have led to tremendous improvements in ML model performance (Lecun, Bengio & Hinton, 2015). As is the case in many fields of science, the use of ML is increasing rapidly in palaeontology and archaeology alike (Bickler, 2021; Fiorucci et al., 2020; Yu et al., 2024). Examples include taxonomical identification (e.g., Moclán et al., 2023; Wills, Underwood & Barrett, 2021), object classification (Resler et al., 2021), object detection (Verschoof-van der Vaart et al., 2020), surface feature extraction such as pottery decoration (Gualandi, Gattiglia & Anichini, 2021) and bone taphonomy (e.g., Courtenay & González-Aguilera, 2020; Moclán et al., 2020; Vegara-Riquelme et al., 2023) as well as datamining for archaeological texts (Brandsen, 2022).

The combined use of CS and ML creates strong synergy in generating large and relevant datasets and analyses that drive new research. The CS-ML approach has seen a strong rise in recent years (Ponti & Seredko, 2022), most notably in the field of biodiversity monitoring (McClure et al., 2020). Combining CS and ML plays to the strengths of both, as it leads to large datasets that can be analysed quickly and accurately. Whereas a CS approach is known to yield serendipitous discoveries (see above), an ML approach can lead to new insights from large scale patterns (McClure et al., 2020). Furthermore, a CS-ML approach elevates public engagement through community platforms and apps such as Observation.org and ObsIdentify (Schermer & Hogeweg, 2018). Such apps and platforms introduce a wide audience to a (research) topic, connects non-experts to experts and lowers the participation threshold for non-experts through fun activities (gamification), feedback and training opportunities. Observations by citizen scientists from these and other platforms such as iNaturalist (2024) are made available to (ML) research through large biodiversity datasets such as GBIF (GBIF.org, 2024). Thus far, examples of a CS-ML approach within palaeontology and archaeology are sparse (e.g., Lambers, Verschoof-van der Vaart & Bourgeois, 2019). These object-based fields have much in common with the observation-based field of biodiversity monitoring but also have some important differences. Palaeontological and especially archaeological collecting is subject to (often) stringent regulations (see below). Another large difference is that collected objects can be treated after collection under better-organised conditions (e.g., standardised photography). This provides some additional considerations to make a CS-ML approach for fossils and artefacts successful in comparison with biodiversity projects.

Fossils and artefacts bring a unique set of challenges and opportunities when considering a CS-ML approach, as palaeontology and archaeology share an object-based character and have similar existing CS networks. Common challenges include studying rare objects which often results in incomplete and sparse data that can stifle ML model performance. Therefore, the need exists for ways to mitigate a lack of data, reduce data biases and increase data quality. The CS-ML approach we review in this study can contribute to those needs. A general scepticism among scientists about the use of CS data (Vohland et al., 2021) and ML-based predictions (Hassija et al., 2024) is increasingly addressed by integrating AI into workflows that include outlier detection and human expertise (e.g., Bourgeois et al., 2024). Moreover, citizen scientists or members from the general public might be sceptical about the use of ML models for highly specialised or niche topics such as the study of fossils and artefacts. High model performance, applicability and an explainable ML model will lead to more trust in the CS-ML approach. A major difference between studying artefacts and fossils are the varying legal contexts, which depend on the country or region in which the objects are found. In some cases, an object is both a fossil and an artefact, such as bone and antler tools made by early modern humans (e.g., Fig. 1, Dekker et al., 2021) which can cause confusion as to what rules apply (e.g., is there an archaeological reporting obligation?). The proposed CS-ML approach will be able to assist collectors and the network of validators and associated scientists to adhere to regulations. It also may form a powerful tool to educate contributors on laws and regulations. Another difference between palaeontological and archaeological objects is the fact that the latter are not as standardisable as taxonomic entities or skeleton elements of fossils. A CS-ML approach can help participants to identify and learn about their objects and become embedded in academic and applied science.

Figure 1 Both a fossil and an artefact.

Mesolithic (8000–4000 B.C.) tool made from red deer antler and wood (U 2014/12.14; National Museum of Antiquities, Leiden, The Netherlands; provenance: North Sea).

In this article, we aim to construct a workflow for object-based palaeontological and archaeological projects involving both CS and ML based on a review of best practices and identified challenges from published studies and projects as well as the authors’ experiences. The proposed workflow can potentially improve existing collaborations and strengthen communities, as well as generate new interest in the topic among a broader range of stakeholders. Moreover, the workflow can be used to monitor finds and observations which can be used by stakeholders such as policymakers, as is already practiced in the field of biodiversity monitoring.

Beach nourishments in the western Netherlands form an ideal case study to explore the object-based CS-ML approach in practice (Fig. 2). For fossils, already since 1874 collaborations have been developed between fishers and the palaeontological community in the Netherlands (Mol, 2016; Mol et al., 2006). The long ongoing collaborations have resulted in major collections and important discoveries, both with institutes and with private collectors. The latter have become more professional over time in both collecting and documenting, becoming a major source of scientific data and materials. The vast volume of Quaternary vertebrate fossils and hominin artefacts that have been collected over the past 150 years (and which continue to be found) form a suitable dataset for ML (Amkreutz & Van der Vaart-Verschoof, 2021; Kuitems et al., 2015; Moerdijk et al., 2010; Mol, 2016; Mol et al., 2006; Mol & Bakker, 2022). The Netherlands fosters a thriving CS community that continues to make new discoveries and communicates findings with the broader public. Importantly, the lack of specific legislation on paleontological heritage in the Netherlands encourages engagement and open collaboration between collectors, museums and scientists (Haug et al., 2020). For hominin artefacts and other ex-situ archaeological finds, systems are in place to report finds by the general public (Ministry of Education, Culture and Science). However, many people are not aware of these systems, resulting in some finds that remain unreported. A CS-ML workflow could help raise awareness about the rules that apply with non-experts who make a discovery. This setting is used to illustrate and discuss the implementation of the CS-ML workflow we propose.

Figure 2 From nourishments to biome reconstruction.

(A) Sediment from the seafloor is “rainbowed” onto the beach (Photo: F.P. Wesselingh). (B) Citizen scientists and enthusiasts search for and (C) find fossils and artefacts (Photos: H.J. Ahrens, Naturalis). (D) Extensive collections (Photo: N. den Ouden, Naturalis) (E) help with reconstructing Late Pleistocene landscapes and biomes (reconstruction by E.J. Bosch, Naturalis).

Methods

A review was conducted with the aim to construct a workflow for palaeontological and archaeological projects involving both CS and ML based on best practises and identified challenges from previous studies and projects. For the purpose of this review, palaeontology and archaeology were grouped together, as there is overlap in (the nature of) studied materials and methods (Figs. 1 and 2). The PRISMA 2020 systematic review approach (Page et al., 2021) was partially adapted to provide a transparent and replicable method which allows other researchers to continue the review based on new literature in the future. Our review method involved systematically searching and screening articles as well as collecting and synthesising data from the screened articles.

The systematic search was conducted through Web of Science (WoS). First, an initial search provided the search strings for each domain to be included in the full systematic literature search (Table 1). Second, the title, abstract and list of keywords were searched for in the included search strings. No further filters or limitations were applied. Because combining the search strings for all three domains yielded few results, three additional searches were conducted. In each search, two out of three domain search strings were combined. The systematic search resulted in a combined 2,083 unique articles (Fig. 3).

Table 1 Search strings for the literature search through WoS for the different domains.

Domain	Search string	
Palaeontology & Archaeology	(paleontolog* OR fossil* OR palaeontolog* OR archeolog* OR archaeolog*) NOT (fuel* OR energy)	
Citizen science	(“citizen science” OR “public engagement” OR “public data” OR amateur*)	
Machine learning	(“artificial intelligence” OR “machine learning” OR “deep learning” OR “computer vision” OR “supervised learning” OR “convolutional neural network” OR CNN* OR “transfer learning” OR “image recognition” OR “image classification”)	
Note:

Asterisks are used for multiple character searching.

Figure 3 Number of results for the literature search through WoS for the different domains.

P, palaeontology; A, archaeology; CS, citizen science; ML, machine learning.

The title and abstract of the search results were screened based on inclusion and exclusion criteria (Table 2) through the online systematic review platform Rayyan (https://www.rayyan.ai/; Ouzzani et al., 2016). Two authors (IE, FW) independently screened a subset of the search results to check for consistency and reproducibility, after which the first author conducted the full screening. In case of doubt concerning an article’s eligibility based on the title and abstract, the full article was screened. Based on the inclusion and exclusion criteria, 66 articles were found eligible. To capture reports and publications not represented by the WoS search and articles published after the search was concluded (April 2023) we consulted all co-authors to identify further relevant articles and reports. This led to the identification of seven additional articles and reports that were screened and found eligible in the same way as those identified through the WoS search.

Table 2 Inclusion and exclusion criteria.

Criteria	Inclusion	Exclusion	
Article type	Peer reviewed articles, working/conference articles, book chapters, reports	Any other type	
Language	English, Dutch	Any other language	
Relevance	Articles on ML projects including both: CS data and/or data attainable by CS (e.g., with a phone camera)

Fossils and artefacts (or objects that share characteristics such as fragmented material, surface modifications and sparse datasets)

	Articles on ML projects including none or one of the relevance inclusion criteria	
Articles on CS addressing either: CS data quality

CS data biases

The role of CS and ML in a project

CS platforms, apps and general public engagement

	Articles on CS addressing none of the relevance inclusion criteria	
Reviews, syntheses or surveys addressing two or more of the following topics: Fossils or artefacts

Citizen science

Machine learning

	Reviews, syntheses or surveys addressing one or none of these topics	
Note:

ML, machine learning; CS, citizen science.

A general characterisation of the screened articles included year of publication, journal and topic keywords (Table S1). Furthermore, we summarised and compared project results, workflows and suggestions for workflows and further research (Table S2). We organised these findings based on different identified aspects of the CS-ML approach: (1) CS data: understanding biases and improving quality; (2) ML models for CS data classification; (3) Roles and task allocation in the CS-ML approach; and (4) CS engagement: platforms, apps and infrastructures. We then synthesised the main findings across categories in an ideal CS-ML workflow. Finally, aspects of the workflow were compared to experiences in the preparation and initial execution phase of our LegaSea project that deals with fossils and artefacts from nourished beaches in the Netherlands.

Results

General characterisation

A general characterisation of the reviewed articles (Table S1) illustrates the novelty and rapid development of the CS-ML approach. The first publication which meets our criteria is from 2005, with the majority published since 2020 (Fig. 4). Furthermore, the reviewed articles were published in a wide variety of journals. The articles are spread thinly over journals; no well-recognized journal exists for the subject.

Figure 4 Year of publication of the included studies.

Note that 2023 only includes studies up to April 2023.

The most frequently used keywords in the reviewed articles (Table 3) largely match our systematic search criteria (e.g., “citizen science”) or fall within the same category (e.g., “crowdsourcing”). However, recurring keywords such as “biodiversity” and “management” do highlight the early successful adoption of the CS-ML approach in biodiversity research and conservation management (e.g., McClure et al., 2020). With research domain differences in mind, the applications, best practices and challenges of CS and ML reported in these articles should be informative for our innovative workflow for a CS-ML approach in palaeontology and archaeology (see Discussion).

Table 3 Topic keywords most frequently used in the reviewed articles.

Topic keyword	Number of occurrences	
Citizen science	31	
Machine learning	16	
Artificial intelligence, classification	12	
Deep learning	9	
Models, learning, archaeology	6	
Biodiversity, identification, image classification, tool, crowdsourcing	5	
Algorithms, humanism, humanities, humans, intelligence, lidar, recognition, remains, conservation, big data	4	
Abundance, computer vision, galaxy zoo, management, nerve net, Netherlands, neural networks (computer), patterns, volunteers, science	3	
Note:

In bold are the keywords that do not directly match (part of) the search strings used in the literature search.

The CS-ML approach: potential and pitfalls

An integrated CS-ML approach offers opportunities beyond the opportunities CS and ML present on their own. CS-ML projects can greatly improve the speed of data acquisition and validation as well as task allocation, allowing real-time action (Green et al., 2020; McClure et al., 2020; Trouille, Lintott & Fortson, 2019). CS-ML data validation can achieve higher accuracy than data validation by CS or ML alone (Green et al., 2020; Lotfian, Ingensand & Brovelli, 2021; McClure et al., 2020). The discovery potential of a dataset is increased with a CS-ML approach due to the complementary serendipitous discovery potential of CS and the complex pattern discovery potential of ML (McClure et al., 2020). With rapidly increasing efforts in collection digitisation and digital data generation in both palaeontology and archaeology, the combined effort of CS and ML can help recognise new/unique objects (e.g., Verschoof-van der Vaart & Lambers, 2021). Furthermore, the CS-ML approach has great public engagement potential (Green et al., 2020; McClure et al., 2020) by generating real-time feedback and promoting interdisciplinary engagement (Lotfian, Ingensand & Brovelli, 2021) (further discussed below). Finally, the potential efficiency of the CS-ML approach presents a cost-effective method when it comes to resources (McClure et al., 2020).

However, the CS-ML approach also poses challenges that need to be addressed. Inaccuracies might arise from low quality data and otherwise biased data (see below). A general lack of trust or scepticism towards CS data leads to a low scientific publication output (McClure et al., 2020). To the same effect, the “black box” problem of ML models (i.e., the inexplicability of the inner workings of the neural networks), makes interpreting results less transparent, unreproducible and hinders engagement (McClure et al., 2020). Engagement can be limited by citizen scientists feeling like the ML model is fully automating their input (Lotfian, Ingensand & Brovelli, 2021). Finally, the threshold for starting a CS-ML project might be high due to the dependency on the large effort of training volunteers, the effort of designing and implementing the project and the required hardware and knowhow (McClure et al., 2020). Moreover, these prerequisites require sufficient financial resources (McClure et al., 2020). These general pros, cons and “lessons learned” for the CS-ML approach, which have emerged over recent years from fields such as biodiversity monitoring, inform our CS-ML approach in palaeontology and archaeology (see Discussion).

CS data: understanding biases and improving quality

To reach the full scientific potential of CS data, biases and varying data quality need to be addressed. CS provides opportunities for large amounts of meaningful data relevant to science that even may help to shape research agendas (Smith, 2014). Contributions by CS include field data acquisition (photography, observations, collection), classifying data, crowdfunding and crowdsourced computer entries (Smith, 2014). The aims and objectives of a research project determine which CS groups to involve and how to engage with contributors (Spyrou et al., 2022). However, a general scepticism and lack of trust towards CS data has been reported (e.g., Bird et al., 2014; Jiménez, Triguero & John, 2019; Santos-Fernandez et al., 2021). This challenge is not limited to CS as biases prevail in datasets annotated by experts (e.g., Bennett et al., 2023; Gooliaff & Hodges, 2018; Quintus, Day & Smith, 2017; Risbøl et al., 2013). Several studies have compared expert and CS data (e.g., Austen et al., 2016; Kallimanis, Panitsa & Dimopoulos, 2017). Non-expert CS data was found to be accurate, but significantly less complete than expert data (Kallimanis, Panitsa & Dimopoulos, 2017). Investigating how experts and citizen scientists make observations should be part of the study design in order to improve CS training and ML model performance (Austen et al., 2016).

Biases and quality metrics need to be identified and assessed before methods for mitigating biases and improving data quality can be incorporated in a project workflow. There are several bias types known in datasets that require manual work in the collection or classification of data. McKay & Kvammen (2020) report on “physical comfort bias, data contrast bias, environment contrast bias, repetition bias, learning bias, feature bias, ambiguity bias and expert knowledge bias”. Furthermore, Koch et al. (2023) discuss a “recognisability bias” where the most recognisable species are more prevalent in the data, creating a bias for both human and ML classifiers. The same bias was observed for expert annotated data (Hsiang et al., 2019). Specific to CS data, Koch et al. (2022) identified a bias towards “attractive” species. Although gamification aspects are incorporated in many CS projects to increase engagement (e.g., Senatore et al., 2020), it has been found to increase bias due to competition and cheating (Piekarczyk et al., 2021). McKay & Kvammen (2020) suggest devising experiments to test for different biases in isolation, such as reversing the order of classification between participants to assess repetition bias and learning bias. Other methods for bias assessments include a model for distinguishing between bias inherent to the whole dataset, a single observation or individual classifier (Jiménez, Triguero & John, 2019) and a model trained on true signals and artificial artefacts to distinguish between the two (Piekarczyk et al., 2021).

A number of methods exist that reduce classification bias (Table 4). Primarily, authors suggest the standardisation of data practices such as “persistent storage, metadata standardization and the other FAIR data principles” (Koch et al., 2022, p. 6). Gooliaff & Hodges (2018) recommend that researchers are explicit about their classification methods and express doubt when unsure about a classification. Taxonomic knowledge and methods of experts should be standardised (Bird et al., 2014) and shared, e.g., in the form of an identification key (Koch et al., 2023). The incorporation of metadata such as spatial data (Santos-Fernandez et al., 2021), classification data (the identity and number of annotators) (Gooliaff & Hodges, 2018) and sampling process metadata (Bird et al., 2014) can further reduce bias. By calculating the “Value of Information”, i.e., the value of additional data per class, Koch et al. (2022) counter sampling biases towards “attractive” species such as birds and butterflies. Moreover, several models are presented to reduce bias such as a spatial Bayesian hierarchical model for dealing with spatial dependence in the data (Santos-Fernandez et al., 2021). Other human biases can be incorporated in ML models such as modelling forgetting (Crowston et al., 2019). This augmentation is an example of mimicking human traits, in this case forgetting a skill if it has not been performed for a certain period of time (Khajah, Lindsey & Mozer, 2016). Bird et al. (2014) present an overview of statistical solutions for CS data biases and a summary of the findings from individual studies related to the implemented models.

Table 4 Overview of methods used to evaluate and reduce biases and improve the quality of CS data.

Study	Methods	Findings	
Bird et al. (2014)	A review of various statistical tools for assessing CS data quality and biases	Collaboratively design the (meta)data collection to allow for characterisation of possible issues with the dataset. Choose the best method to deal with those issues.	
Shamir, Diamond & Wallin (2016)	Calculating annotation consistency	Reduces required number of annotations, improves overall quality of data and can be used to generate real time feedback.	
Salk et al. (2017)	Inter-volunteer agreement to evaluate task difficulty	>90% for 80% of the data, negatively correlated with expressed uncertainty. >20% misclassification if only volunteers’ majority vote was used.	
Gooliaff & Hodges (2018)	Various statistical measures for expert agreement	Classifications by a single expert were somewhat unreliable (87% match with majority classification), but final majority classifications were correct most of the time (no rate because no ground truth classifications).	
O’Leary et al. (2018)	Statistical procedure to divide tasks for experts and non-experts	>90% correctly scored data while reducing workload for experts by 50%.	
Soul et al. (2018)	Various statistical methods to reduce CS inaccuracies	CS undercount epidermal cell counts relative to experts in 92% of cases, showing no improvement with experience. Key lessons: acknowledge uncertainty, accommodate various levels of expertise, provide examples and annotated graphics.	
Crowston et al. (2019)	Bayesian model for tracking volunteer learning	Comparing ML classification confidence and human-ML agreement allows to empirically divide the dataset into a training set for beginners, a set to be classified by trained volunteers and a set of rare/difficult and unknown classes. Volunteers of different levels are trained while all advance the project’s work.	
Jiménez, Triguero & John (2019)	Transformations and hybridisation to aggregate transformed data to improve quality	The Galaxy Zoo case study showed greater accuracy and fewer unclassified images, showing the approach is “able to find an adequate adjustment for the aggregation of information about the uncertainty present in the data”.	
Langenkämper et al. (2019)	Modifying “gold standard” annotations to resemble CS data	Training a CNN on simulated data reduces the effect of inaccurately labelled CS data, which should also be reduced by organising an in-person training session or by limiting the number of classes.	
Brenskelle et al. (2020)	Test annotation accuracy in in-person and online settings	In-person annotators scored consistently high (90–100%), whereas the online volunteers varied widely in their accuracy (65–100%). Volunteers who scored more images, performed better. Triplicate scoring outperformed single scoring by ~3%.	
McKay & Kvammen (2020)	Varying ergonomic circumstances during the classification process	Four key human traits that affect classifications: short-term memory limit, boredom and fatigue, recency effect and positivity bias. Physical discomfort reduced performance.	
Santos-Fernandez et al. (2021)	Bayesian model to learn about coral reef cover	More precise regression coefficient estimates, while accounting for varying participant abilities.	
Mugford et al. (2021)	Bayesian model for calculating classification certainty	>60% of observations that are not yet considered research grade could be upgraded based on classification certainty, optimising the use of CS data and efforts.	
Pataki et al. (2021)	Deep learning model trained on subsets of data to explore data quality improvements	Greater class imbalance (94% vs 70–80%) led to poorer model performance (ROC AUC score of 0.83–0.84 vs 0.90–0.93).	
Piekarczyk et al. (2021)	Convolutional neural network (CNN) trained on distinguishing true signals and artefacts	Almost 99% recognition ratio for both signal and artefacts, allowing elimination of manual supervision of the competition process.	
Koch et al. (2022)	Value of information (VoI) of added data for an image recognition model	The VoI of underrepresented orders is generally higher, but the expected increase in performance with more data differs per order, allowing a better informed data collecting strategy.	
Koch et al. (2023)	CNNs and comparing F1 scores with data availability	Strong correlation between high data availability and model performance for birds and most other considered orders (all classes had the same amount of training data). No apparent correlation for beetles and lichens.	
Perez-Udell, Udell & Chang (2023)	K-means clustering to aggregate like-colour pixels	Three reviewers agreed unanimously on 72.6% of colour classifications which in turn had a 85.3% agreement with algorithm classifications.	
Bourgeois et al. (2024)	Inter-user agreement and (validating) fieldwork	With an inter-user agreement of 12+ the precision increases above 0.82 (and to 0.7 for an inter-user agreement of 7+), arguably doubling the number of known burrows in a cost effective way.	

To address frequent concerns about CS data quality such as poor image quality and misclassifications by non-experts (Mugford et al., 2021; O’Leary et al., 2018; Pataki et al., 2021; Bourgeois et al., 2024), the quality of CS data can be assessed and improved. The quality of CS datasets can be assessed by acknowledging and addressing uncertainty (Soul et al., 2018) and by designing “task difficulty metrics” based on the uncertainty and disagreements among participants (Salk et al., 2017). It can also be assessed by measuring the usefulness of data for classification tasks (Pataki et al., 2021). Crowston et al. (2019) suggest comparing the performance of untrained and trained volunteers. For trained volunteers the performance of those trained online and those trained in person can be compared (Soul et al., 2018).

To improve the quality of CS datasets, ML and other statistical models are used in a variety of ways (Table 4). For annotation tasks, ML models and other statistical models can present a citizen scientist with data to annotate based on experience (Crowston et al., 2019; O’Leary et al., 2018; Pataki et al., 2021; Soul et al., 2018) and present the user with tailored real-time feedback (Shamir, Diamond & Wallin, 2016). O’Leary et al. (2018) found that the workload for experts could be reduced by 50% while maintaining >90% correctly scored data. The experience of an individual can also be used to weigh their input, reducing the number of annotations needed per observation (Mugford et al., 2021; Shamir, Diamond & Wallin, 2016) and upgrading >60% of CS data to research grade data (Mugford et al., 2021). Moreover, Langenkämper et al. (2019) use a ML model to artificially alter a “golden standard” dataset to better resemble a CS dataset which in turn is used to train a ML image identification model. Perez-Udell, Udell & Chang (2023) designed a ML model that classifies CS images with varying lighting conditions based on the Hue-Saturation-Value instead of RGB colour values and found an 85.3% agreement with unanimous reviewer agreement. Furthermore, a tutorial can help volunteers to prepare for the task at hand (Brenskelle et al., 2020; Salk et al., 2017). Participants are provided with clear instructions for the best ways to take pictures such as camera angle and light conditions (Perez-Udell, Udell & Chang, 2023) and they are provided with examples of suitable, expert-validated data and examples of attributes expert classifiers look at through annotated graphics (Salk et al., 2017; Soul et al., 2018). Through message boards, participants can exchange experiences and ask each other and experts questions, increasing motivation and self-efficacy (Soul et al., 2018; Bourgeois et al., 2024). Clear language will prevent misinterpretation of the task (Salk et al., 2017). Brenskelle et al. (2020) found that in-person training yields consistently higher accuracy scores (90–100%) than online training (65–100%). Online training can be more time-efficient, however (Brenskelle et al., 2020; Langenkämper et al., 2019). Finally, by incorporating user feedback in the tutorial or training phase, CS data quality can continuously be improved (Soul et al., 2018).

ML models for CS data classification

The majority of reviewed studies use convolutional neural networks (CNNs), which are the current industry standard ML method for image classification tasks (Wang, Fan & Wang, 2021), although quickly being replaced by Transformer-based architectures (Dosovitskiy et al., 2020). Image classification tasks feature heavily among the screened articles because CS is easily incorporated in such projects. Citizen scientists can collect image data with their phone and/or annotate pictures through an online platform with limited knowhow. Data that was used in the reviewed projects originate from different sources such as CS platforms (e.g., Perez-Udell, Udell & Chang, 2023; Suzuki-Ohno et al., 2022), online repositories (e.g., Lambers, Verschoof-van der Vaart & Bourgeois, 2019; Resler et al., 2021) and public data assembled using web-crawls (Liu et al., 2023).

The reviewed articles include a myriad of applications for the ML approach such as identification or classification (e.g., Flores et al., 2019; Gualandi, Gattiglia & Anichini, 2021; Liu et al., 2023), object detection (e.g., Verschoof-van der Vaart et al., 2020; Verschoof-van der Vaart, 2022) and grouping of objects (Parisotto et al., 2022; Resler et al., 2021). Species identification or other systematic or hierarchical grouping of objects or data is often imprecise and subjective (Matthews et al., 2018) or ambiguous (Resler et al., 2021), even if performed by experts. In some cases, ML methods have been shown to be more consistent and to outperform experts (e.g., Pawlowicz & Downum (2021) experts: 73.6–86.9%, ML: 82.5%; Resler et al., 2021 experts: 20.6–44.4%, ML: 69.8%), although with more complex tasks this is often not yet the case (Verschoof-van der Vaart, 2022). Furthermore, ML methods can reduce time and effort when exploring or analysing large datasets (Perez-Udell, Udell & Chang, 2023) and reveal yet unknown patterns across a dataset (Resler et al., 2021).

A number of studies identify specific applications of ML methods in palaeontology and archaeology (Table 5). ML models can be used for automatic fossil and artefact identification (e.g., Chalechale, Bahri & Vatanchian, 2010) as well as other computational tasks such as (3D) digitisation of collections/data and faster searching of texts/reports (Boon et al., 2009). The scalability, flexibility and rapid development of ML models can help us understand complex relations between organisms, their environments and their evolution (Žliobaitė et al., 2017; Bickler, 2021). However, datasets of fossils and artefacts are often relatively small. Therefore, ML models might not identify hidden complexities and identify systemic biases as true patterns (Bickler, 2021). The accumulation of data and the validation and analysis of datasets, including metadata, is therefore essential (Fiorucci et al., 2020), but time consuming and expensive (Bickler, 2021). To train robust models based on high quality data requires a strong collaboration between computer scientists, palaeontologists and/or archaeologists (Boon et al., 2009).

Table 5 Overview of reviewed ML model solutions for analysing CS data (or similar) in palaeontology and archaeology and neighbouring fields with similar approaches such as sparse data augmentation, metadata inclusion and colour analysis.

Study	Problem definition	Strategy	Findings	
Flores et al. (2019)	Projectile point classification is an extensive and complex process	Two part algorithm: projectile point component and global shape. Pre-segmentation allows for comparing variated datasets	Algorithm evaluation shows an "acceptable" F1-score of 0.7. Main points of improvement: metadata inclusion.	
Lambers, Verschoof-van der Vaart & Bourgeois (2019)	Lack of suitable archaeological training data for automated object detection	Combining ML-based object detection, CS-based data annotation and validation and a CS fieldwork campaign	Zooniverse project ‘heritage quest’ and ML workflow WODAN.	
Subirats et al. (2019)	Teaching students complex tasks in the field involving large amounts of data	Tutoring system that helps student classify bone fragments	Random forest method resulted in F1-score between 0.83–0.86 and lists of characteristics that students can use for identifications.	
Terry, Roy & August (2020)	Accurate species identification from CS images	Multi-input NN that includes metadata	9.1% increase for multi-input model vs 3.6% increase for an ensemble image and metadata model against a image-only baseline of 48.2%.	
Verschoof-van der Vaart et al. (2020)	False positives caused by specific regions in the research area	Metadata inclusion: location-based ranking incorporated in pipeline	F1-score increase from 46–50% to 70% for detecting Celtic fields and barrows.	
Anichini et al. (2021)/Gualandi, Gattiglia & Anichini (2021)	Ceramics classification is time consuming, manual work and sources are not widely accessible	Two complimentary CNNs, synthetic training data generation	Potsherd identification tool for fieldwork and post-excavation analysis. Although accuracies are not very high (19% and 55% top-1 accuracy), still useful to professionals.	
De Lutio et al. (2021)	Classification based on images alone is challenging and classes are imbalanced	Metadata inclusion: spatial, temporal and ecological context, hierarchical taxonomic structure	CNN that takes metadata into account outperforms CNN trained on image data alone (79% vs 73% accuracy).	
Pataki et al. (2021)	Limited data availability due to labour-intensive process	CS system with CNN identifications	High performance, which was further improved with class balancing (ROC-AUC from 0.9577 to 0.9663).	
Pawlowicz & Downum (2021)	More direct comparisons needed for deeper understanding (e.g., time relationships, stylistic trends)	CNN models, compared to human and consensus results	CNN models perform similar to expert annotators (accuracy of 82.5% vs 73.6–86.9%) and help identify features for reliable classification by depicting salient attributes as heat maps.	
Resler et al. (2021)	Pattern discovery in large datasets	CNN model and community detection algorithm based on confusion matrix	CNN model had comparable performance as experts in their field of expertise, but outperformed them in other classes with average accuracies of 69.8% vs 20.6–44.4%.	
Verschoof-van der Vaart & Landauer (2021)	Large amount of available high-quality data to analyse	Employ CNNs in computationally effective pipeline	Using digital terrain model data in combination with location-based ranking outperformed visualised data (F1-score 0.50 vs 0.44).	
Verschoof-van der Vaart & Lambers (2021)	Need to test models on data from different context	Testing model on data from a different area in the Netherlands	A combined human-computer strategy can offset inherent biases in manual analysis yet detect objects from a wide variety of target classes.	
Lengauer et al. (2022)	Lack of training data for geometric ornamentation classification	Annotation system that feeds input to a CNN	“The generated annotations exhibit a high degree of structure with references to specific surface areas and digital links to similar patterns”. Moreover, the workflow can be generalised to a wide range of domains.	
Parisotto et al. (2022)	The large variation in potsherds makes classification challenging	A deep convolutional variational autoencoder (vae) network for potsherd pairing	“Features are proved to be robust to a number of perturbations mimicking the real scenario of the high shape variance due to handcrafted materials”.	
Wei (2022)	Pottery identification is time consuming and costly	Mask r-CNN model to enhance features of the outer contour of the pottery	This method outperforms other ML methods for all different examined categories of pottery with a general accuracy above 90% (87% for pattern decoration pottery).	
Liu et al. (2023)	Taxonomic identification of fossils is often labour-intensive, tedious and requires expert knowledge	Train CNNs on a web-crawled dataset	High average accuracy of 90%, F1-score varies from 0.99–0.55, most being in the high 0.80 and 0.90 s.	
Perez-Udell, Udell & Chang (2023)	Difficult to retrieve reliable colour data from images (relevant for e.g., taphonomy and pottery decorations)	K-means clustering to aggregate like-colour pixels	Three reviewers agreed unanimously on 72.6% of colour classifications which in turn had a 85.3% agreement with algorithm classifications.	

Even though the reviewed articles represent a variety of workflows and methods, they contain common successful approaches and best practices in ML projects useful for a workflow design. In successful projects, academics, data scientists and citizen scientists are involved in the definition of the research project (Lambers, Verschoof-van der Vaart & Bourgeois, 2019) to get the most out of the project for everybody involved. To answer the relevant research questions using ML models, large and high-quality datasets are required. Typically, the distribution of data among the different classes is imbalanced which results in classes being over-/underrepresented in the dataset (Gualandi, Gattiglia & Anichini, 2021) (see above). Furthermore, data from one class can be very diverse (intraspecific variation) while data from different classes can be similar (interspecific similarity) (Gualandi, Gattiglia & Anichini, 2021). Examples of creating larger and more diverse datasets include synthetic data generation (e.g., Anichini et al., 2021; Gualandi, Gattiglia & Anichini, 2021) and data augmentation (e.g., Wang et al., 2022). However, the training datasets and especially the test data should be a reflection of the data for which the ML models are eventually used, i.e., real-world situations (Verschoof-van der Vaart, 2022). Classification is further improved by incorporating metadata such as date, location, altitude and user ID (e.g., De Lutio et al., 2021 +6%; Terry, Roy & August, 2020 + 9.1%) and a hierarchical structure (e.g., De Lutio et al., 2021) in the dataset. Additionally, ML models can be tweaked for dealing with small datasets, for example by using Transpose CNNs (Wang et al., 2022) or by adapting the loss function (Wei, 2022). Also, ML models can be tweaked for dealing with overfitting due to unbalanced classes, for example using a late fusion strategy (De Lutio et al., 2021), a large dropout (Anichini et al., 2021), or class balancing strategies (Johnson & Khoshgoftaar, 2019). ‘Heat maps’ (Liu et al., 2023; Pawlowicz & Downum, 2021), ‘ROC AUC scores’ (Suzuki-Ohno et al., 2022), ‘MCC scores’ (Verschoof-van der Vaart & Landauer, 2021) and ‘confusion matrices’ (Suzuki-Ohno et al., 2022) are examples of methods and metrics that can be used to evaluate ML model performance. These methods also help address the “black box” of decision making (how or why a ML model came to a conclusion), a general weakness of ML models (Pawlowicz & Downum, 2021). Finally, making the ML model available to the public increases data generation and public engagement (Flores et al., 2019; Liu et al., 2023; Parisotto et al., 2022).

Many of the studies identified challenges and additional potential improvements that are relevant for an optimal CS-ML workflow. First, in almost all articles the authors state the need for expanding the dataset and the need for a more diverse dataset. Gualandi, Gattiglia & Anichini (2021) also specifically advocate the use of FAIR (Findable, Accessible, Interoperable and Reusable) data (Wilkinson et al., 2016). The development of public online platforms and apps can help with data generation, adherent to the FAIR principles (see below). ML methods can further be employed on these CS platforms for task allocation and data quality evaluation (see above). Second, as ML methods are constantly being improved upon, larger, faster and otherwise improved models are proposed (Anwar, Sabetghadam & Bell, 2020; Pawlowicz & Downum, 2021; Suzuki-Ohno et al., 2022). ML models could also be used in conjunction as ensemble models (Terry, Roy & August, 2020). For application in CS projects in an uncontrolled environment, the robustness against outliers (images with unknown classes, invalid input, etc.,) is an important property (Ruff et al., 2021). In automated environments, where the ML model is being used to automatically label part of the data without human supervision, it is important that the predicted probabilities are properly calibrated (Nixon et al., 2019). Finally, in order for a CS-ML project to be successful and to increase its longevity, there is a need for regular feedback and support for citizen scientists before, during and after the duration of the project (Lambers, Verschoof-van der Vaart & Bourgeois, 2019). Examples include a user study (Lengauer et al., 2022) and providing training and information (Lambers, Verschoof-van der Vaart & Bourgeois, 2019). Attention for the CS participants and maintaining relations takes huge efforts and the ML methods described here may help to alleviate some of this pressure (see Discussion).

Roles and task allocation in the CS-ML approach

The appropriate definition of roles and tasks, and management of task allocation in a complex CS-ML project with a hugely varied participant field determines the success of such projects. The reviewed CS-ML projects involve similar recurring tasks for different stakeholder and participant groups and clarifying these tasks at the onset of a project helps clarifying the goals of the project. Task allocation in CS-ML projects can be based on the most efficient way of attaining high accuracy data for scientific output and based on maximising CS engagement. Both are valid options and a balance can be struck depending on the project at hand. Ponti & Seredko (2022) identify tasks for citizen scientists, experts and ML models. CS tasks require common skills and have a low level of interdependence such as data collection and classification tasks (Ponti & Seredko, 2022). Expert tasks require expert skills and a medium to high level of interdependence such as CS data classification, teaching citizen scientists, creating “gold standard” datasets and data analysis (Ponti & Seredko, 2022). ML model tasks have a high level of interdependence such as data classification, data analysis and evaluation of CS data quality (Ponti & Seredko, 2022). CS can be used to validate ML classifications and ML can be used to validate CS classifications (although it is important to avoid circular reasoning in a workflow). Also, a consensus can be reached between CS and ML models. ML models can further be used to filter out more difficult classifications for expert validation (Green et al., 2020). This way, task allocation determines what knowledge is incorporated in the classification and validation process (Ponti, Kasperowski & Gander, 2022) and addresses the concerns regarding the CS-ML approach in palaeontology and archaeology discussed in this review.

Moreover, the roles and task allocation in a CS-ML project can evolve throughout the duration of a project (Ponti, Kasperowski & Gander, 2022). Green et al. (2020) provide considerations for the roles of CS in a ML project. For the project aims, aspects of research, engagement, education and social benefits can be included. Citizen scientists can contribute to the formulation of research questions for case studies. Methodologies for research and engagement have to lead to meaningful results for both. Besides these considerations that arose from reviewing successful CS-ML projects, it is also essential to include findings from unpublished work and unsuccessful projects (Ponti & Seredko, 2022) by conducting interviews with project contributors (Ponti, Kasperowski & Gander, 2022). In the case of palaeontology and archaeology, where examples of CS-ML projects are scarce, interviewing contributors from neighbouring fields such as biodiversity monitoring may yield useful insights.

Further considerations for the roles and task allocation in CS-ML projects include balancing scientific efficiency on the one hand and CS inclusivity and engagement on the other (Trouille, Lintott & Fortson, 2019). Perhaps not all that can be automated, should be automated. Many fossil and artefact collectors take great joy in the time spent on their hobby and might be discouraged by involvement in a very efficient workflow. Ponti & Seredko (2022) discuss whether automation such as ML serves all citizen scientists, or only those that are familiar with the latest technological advances. CS inclusivity can be improved by incorporating a discussion forum on the CS platform, as this encourages participation, social community building and serendipitous discovery (e.g., Heritage Quest (2024), Trouille, Lintott & Fortson (2019). Furthermore, clear instructions and reward systems can improve participation and data quality (Kelling et al., 2013). Through interviews with stakeholders, Adam et al. (2021) identified additional ways to motivate citizen scientists. These include showing an overview of the data that has been contributed such as statistical overviews, map projections, archiving records and allowing data exports. A major issue in our experience that did not receive much attention in the reviewed literature is the efforts it requires to build, support and maintain communities to stimulate participants to invest time, energy and resources (see also Discussion below). Task allocation among citizen scientists, researchers and other stakeholders can be summarised in a use case diagram for both the research project and the CS platform (Adam et al., 2021; see Discussion). In the next section, these use case diagrams and workflows of CS platforms and apps are discussed.

CS engagement: platforms, apps and infrastructures

The design of CS platforms and apps and the underlying infrastructure is key to Open Science (UNESCO, 2022), to maximising engagement and to ensure durable use and consistent, quality controlled, FAIR data. As is described in the previous sections, making a ML model available to the citizen scientists increases data acquisition and public engagement (e.g., Liu et al., 2023). Moreover, using ML for automatic feedback to citizen scientists improves data quality and participation (e.g., Pataki et al., 2021). A number of reviewed articles report on online CS platforms, apps and other forms of public engagement, which all incorporate some of the findings from previous sections in their workflow. Here, we report on existing use case diagrams and workflows for online CS platforms and apps as well as suggestions for future workflows that are applicable to object-based CS-ML projects.

Generally, an online CS platform comprises three main components: a mobile or web application for the general public (data collection) and volunteer experts (data annotation/validation); a web portal designed for researchers, collection managers and the general public to search the database; and a linked open data service for custom data analyses by data scientists and software developers. In many projects, the mobile or web application can be used by citizen scientists to collect data in the field, e.g., at an excavation site. Apps can include a multitude of features and web portals are designed for researchers, collection managers and the general public to search collection and reference databases (Table 6). The platform presented by Saykol et al. (2005) allows for visual content-based queries aside from semantic searches. Finally, a Linked Open Data service can be designed for custom data analyses by data scientists and software developers (Hyvönen et al., 2021; Jones, 2020). For the collection, storage and use of data on a CS platform or app, the importance of adhering to the FAIR and Linked Open Data guiding principles is emphasised (Carpentier, 2022; Hyvönen et al., 2021).

Table 6 Overview of the main features of reviewed CS platforms and apps.

Reference	Platform/app	ML classification tool	Metadata captured	Expert validation	Collection database	Reference database	Subject info/ educational material	Linked open data service	Login	Offline data collection	Free/paid	
Robinson et al. (2017)	Plankton Portal (Zooniverse)	No	Yes	No	Yes	Yes	Yes	No	Optional	No	Free	
Kress et al. (2018)	Leafsnap	Yes	Yes	No	Yes	Yes	Yes	No	No	No	Free	
Lambers, Verschoof-van der Vaart & Bourgeois (2019)/Verschoof-van der Vaart et al. (2020)/Bourgeois et al. (2024)	Heritage Quest (Zooniverse)	No	Yes	Yes	Yes	No	Yes	No	Optional	No	Free	
Boho et al. (2020)	Flora Capture	Yes	Yes	Yes	Yes	Yes	Yes	No	Optional	Yes	Free	
Bonnet et al. (2020)	Pl@ntnet	Yes	Yes	Yes	YeS	Yes	Yes	No	Optional	Yes	Free	
Senatore et al. (2020)	HeGo project	No	Yes	No	No	No	No	No	No	No	Free	
Anichini et al. (2021)/Gualandi, Gattiglia & Anichini (2021)	ArchAIDE	Yes	Yes	No	Yes	Yes	Yes	Yes	Optional	Yes	Free	
Hyvönen et al. (2021)	FindSampo	No	Yes	Yes	Yes	Yes	No	Yes	Yes	No	Free	
Mäder et al. (2021)	Flora Incognita	Yes	Yes	Yes	Yes	Yes	Yes	No	Optional	Yes	Free	
Pataki et al. (2021)	Mosquito Alert	No	Yes	Yes	Yes	Yes	Yes	No	Yes	Yes	Free	
Carpentier (2022)	Portable Antiquities of the Netherlands	No	Yes	Yes	Yes	Yes	Yes	No	Yes	Yes	Free	
Picek et al. (2022)	FungiVision	Yes	Yes	Yes	Yes	No	Yes	No	Yes	No	Free	
Liu et al. (2023)	ai-fossil.com	Yes	Yes	No	Yes	No	No	No	Yes	No	Free	

Training events and regular feedback improve engagement with public participants and improve data quality. It is considered best practice to organise regular training events for new users to learn about the use of the platform and to collect or validate data (Pataki et al., 2021; Picek et al., 2022; Robinson et al., 2017). Online training and data collecting can also be accompanied by ML models such as natural language models and gamification techniques (Boho et al., 2020; Picek et al., 2022; Subirats et al., 2019). Information can be presented for different target audiences such as children and students (e.g., Heritage Quest Junior, 2024; Mäder et al., 2021). Maintaining frequent communication with participants regarding goals, updates and outcomes of the project will keep them engaged (Bonnet et al., 2020; Robinson et al., 2017). To ensure inclusive and community-based CS, it is recommended to embed open participation from the outset and take portable and ethical science principles in mind (Liebenberg et al., 2017; Milek, 2018). A user-friendly interface with simple icons and clearly defined tasks reduces the participation threshold (Liebenberg et al., 2017; Stewart, Labrèche & Lombraña González, 2020; Bourgeois et al., 2024). Platform or app design might differ for different projects. Therefore, project managers should consult potential users on the design of the online CS platform (Boho et al., 2020; Bonnet et al., 2020; Gualandi, Gattiglia & Anichini, 2021).

Discussion

The reviewed articles provide broad insights into all aspects relevant to the design of a phased workflow for a successful CS-ML approach in palaeontology and archaeology (Figs. 5–8). Standardised workflows and guidelines for projects in a new interdisciplinary field such as discussed here, accelerate the process of starting new projects and help to avoid common pitfalls (Liew et al., 2017). Therefore, the aim of this discussion is to outline a workflow for CS-ML projects based on recommendations and challenges identified in the reviewed literature. We summarise and structure the results of the review in a general workflow overview of a phased CS-ML approach (Fig. 5), the main objectives and challenges of the different domains of a phased CS-ML approach (Fig. 6), the roles and task allocation in a CS-ML project (Fig. 7) and a general infrastructure plan for an online CS platform with ML tools (Fig. 8). Insights from the authors as well as additional experiences from unsuccessful and ongoing (research) projects are included here as well. Moreover, to specifically address the additional challenges provided by object-based workflows compared to the more observation-based biodiversity practices, we discuss examples from the LegaSea project in which fossil- and artefact-rich beach nourishments in the western Netherlands are studied.

Figure 5 Overview of phased CS-ML workflow with the main components.

Figure 6 Objectives (O) and challenges (C) for each of the relevant subject domains in each of the CS-ML workflow phases.

Figure 7 Roles and task allocation in phased CS-ML approach.

Figure 8 App and infrastructure workflow for an online CS platform in a CS-ML project.

The coloured circles within each of the aspects of the infrastructure refer to the target users within different subject domains.

Phased CS-ML approach

The proposed CS-ML workflow consists of four phases: (I) preparation, (II) execution, (III) implementation and (IV) reiteration phase (Fig. 5). Each phase has its own objectives, challenges, milestones and evaluation moments/criteria. In the preparation phase (I) the objective is to establish an outline of the CS-ML project. A major challenge is to establish support (e.g., institutional embedding, financial support and community support) for the duration of the project and the implementation afterwards. As CS-ML projects inevitably require communication over very different domains and with different participants, an advisory board with members from the different domains might be useful for the project manager to strengthen communications. With a well-supported project outline, the preparation phase can be concluded with an evaluation in which possible concerns from contributors and other stakeholders are discussed and incorporated in the next phase. The execution phase (II) (Fig. 5) consists of three parts that are enacted simultaneously: (IIa) data, (IIb) ML model and (IIc) app/infrastructure. The main objective is creating high quality datasets and ML models in an iterative process to identify the ideal data collecting strategy as well as the best performing model architecture and parameters. These strategies feed into the design of the app and infrastructure for public engagement. A major challenge is to reduce biases and ensure data quality. The execution phase is concluded when the online CS platform is ready to launch. The experience of the contributors and the performance of the ML models can then be evaluated. The main objective of the implementation phase (III) is to generate new data and use these for research and outreach. Finally, the reiteration phase (IV), when new data and feedback are periodically used to retrain and update ML models, apps and infrastructure, runs parallel to the implementation phase. The major challenge for both implementation and reiteration phases is the sustained use and support by contributors and other stakeholders, even beyond the project duration. Periodic milestones and evaluations can be based on the scientific output and use of the CS platform.

CS-ML approach objectives and challenges

Within the CS-ML approach we have identified further objectives and challenges that can be organized in five subject domains: CS tasks, ML development, research, stakeholder engagement and app development. Each subject domain has different objectives and challenges that evolve during the successive phases of the CS-ML approach (Fig. 6).

The main objectives of the CS tasks subject domain, are the sustained participation (beyond duration of the project), motivation and self-efficacy of citizen scientists and involve both tasks performed by different CS groups as well as the related organisational and managerial tasks. The main objective during the preparation phase (I) is to outline the CS participation and to initiate CS engagement. By already involving citizen scientists in the planning phase, project managers can incorporate their wants and needs in the design of the project. A main challenge here is to address potential scepticism related to the use of ML and the role of CS in scientific research. During the execution phase (II), citizen scientists are involved in collecting and annotating data, in identifying case studies and in validating ML model predictions. It is important to realise that these tasks are time consuming and that it is a challenge to properly mitigate data biases and varying data quality (e.g., Koch et al., 2022). The objectives related to these challenges are for citizen scientists to provide feedback and to express doubt while performing tasks as well as to create a CS platform that is user friendly and designed for maximum data quality through clear instructions and opportunities for feedback. These can include instructions and feedback on how to orientate the object and how to take pictures for example. To motivate citizen scientists, it should be clearly communicated that their work contributes to the development of ML models and that the citizen science community is properly attributed when releasing models or publishing work in which the models are used. During the implementation phase (III), the main CS task objective is to provide training and information. The challenge is to initiate and maintain engagement with all potential CS groups by increasing motivation and self-efficacy (e.g., Soul et al., 2018; Bourgeois et al., 2024). From our experiences with palaeontological and archaeological collectors, building and maintaining a thriving CS community requires large time investment, attention and recurring participation meetings. Whenever a CS community becomes larger inevitably collaboration issues will arise. A liaison officer role is required to organise meetings, notice signs of discomfort among participants and address these in a timely manner. These responsibilities can also include handling social media accounts related to the project. In the reiteration phase (IV), the main objective is to incorporate user feedback in the continued development of the CS platform by which motivation and self-efficacy can be increased.

For the second subject domain, the ML development, the objective is to deploy a ML model trained on CS data that can be used by the general public. In the preparation phase (I), the objective is to explore the existing ML tools and hardware that are available and to decide on using or converting an existing ML pipeline or to build a new ML pipeline. Sufficient institutional support with regards to finances, resources and knowhow is key. During the execution phase (II), the collected dataset can be made more robust with class balancing, data augmentation or data synthesis as palaeontological and archaeological data are generally scarce (e.g., Bickler, 2021). This way biases can be reduced, and varying quality of the data can be improved. To do so requires domain knowledge and a strong collaboration between computer scientists and domain experts, i.e., palaeontologists and archaeologists. ML models are trained, tested and model performance is evaluated which provides insights into missing or weak parts of the dataset. Both the ML models and the dataset can be improved in an iterative process. Incorporating metadata in the training process can further reduce biases (e.g., Bird et al., 2014; Gooliaff & Hodges, 2018; Santos-Fernandez et al., 2021). A challenge here is to standardise the metadata and to incorporate the metadata collecting in the data collection strategy. Aside from the main ML models, additional models can be trained for task allocation and for providing real-time feedback for the CS platform users. Incorporating these models in the app and infrastructure can further reduce biases and increase data quality. In the implementation phase (III), the objective is to provide live ML models that can be used in the app. A major challenge is to explain the inner workings of the model to users (e.g., Hassija et al., 2024). In the reiteration phase (IV), the model is also analysed, looking for gaps in the dataset and looking for innovative methods to improve model performance. As ML methods develop rapidly, it is a key challenge to keep up with these innovations (i.e., keeping code up to date and making sure dependencies are working) and periodically rebuild the ML models and infrastructure to guarantee compatibility and state-of-the-art model performance.

In the third subject domain, research, the overall objective is new scientific discoveries and insights. Additionally, domain knowledge support for the CS and ML tasks as well as outreach are important objectives. During the preparation phase (I), the specific research objectives are established. The challenge here is to evaluate if the CS-ML approach actually suits these research objectives. This includes an evaluation of the suitability of CS involvement in, e.g., fieldwork with regards to the applicable rules and regulations. In the execution phase (II), research data tasks focus on training citizen scientists, standardising (meta)data collection and annotation and identifying case studies. The major challenge is again to deal with biases and varying data quality. Together, these tasks are time consuming and need to be planned out ahead of time. Validation of ML model predictions focusses on more rare and complex cases. Standardised naming conventions, i.e., thesauri, need to be employed for consistency. Also, for research purposes it is challenging but key to express doubt while performing these tasks to reduce biases in the data and ML models. Additional subject information can be provided for the app for outreach purposes, as well as input on the design of a digital reference collection and CS collection registration system. This is done with input from different user groups in mind. With the app and infrastructure up and running in the implementation phase (III), the main objective of research becomes new discoveries and insights resulting from the project. A challenge here is to fully explain interpretations from the CS-ML approach with its possible biases and to generate scientific output. In the reiteration phase (IV), gaps in the dataset are identified and new sources of data must be found and incorporated in the dataset.

The fourth subject domain, general stakeholder engagement, aims at engaging in a wider societal discourse and to incite sustained real-time (inter-)action. For this purpose, the relevant stakeholders and what they might want to get out of the project need to be identified in the preparation phase (I). An array of approaches for stakeholder identification and engagement exists, e.g., The BiodivERsA Stakeholder Engagement Handbook (Durham et al., 2014). In the execution phase (II), the main objective is to create a FAIR data infrastructure to make sure the data can be (re)used for different projects beyond the scope or topic of the original project. CS stakeholders that invested time in creating/validating datasets should be acknowledged. The implementation phase (III) includes the development of educational material and different outreach packages. Although challenging, a FAIR data infrastructure can incite real-time action in and beyond the scope of the CS-ML project. In the reiteration phase (IV), feedback from stakeholders can be incorporated in the data management and in improving the platform design. Furthermore, additional (meta)data can be collected if this can improve (interdisciplinary) engagement and research.

The objective for the fifth subject domain, app and infrastructure development, is a durable and adaptable infrastructure with an app that is user friendly and leads to high quality, FAIR data suitable for a wide variety of scientific research. During the preparation phase (I), the objective is to explore the requirements of the infrastructure to host the FAIR data, ML models and engagement with researchers and other stakeholders. For such an infrastructure to be durable, sufficient institutional embedding is required. During the execution phase (II), the app and infrastructure are built. A possible challenge that might arise are contradicting infrastructure requirements for the aims of different stakeholder groups (e.g., with regards to privacy and licencing of CS data). In the implementation phase (III), the main objective is to maintain the app and infrastructure, which again requires sufficient institutional embedding. Finally, in the reiteration phase (IV) the aim is to identify and design new features for the app and update the infrastructure when needed.

Roles and task allocation

As CS-ML projects are inherently complex, and involve many actors, a clear definition of roles and tasks is paramount to a successful collaboration. Here, we formulate tasks based on the identified objectives and challenges and allocate the tasks to contributors in different roles (Fig. 7). Tasks can be assigned to multiple roles, individual contributors can perform multiple roles and multiple contributors can in turn perform a single role. Based on the results of our review, combined with the experience of the authors, we propose six categories of contributors/participants, viz. user/contributor, volunteer expert, citizen and professional researcher, computer scientist, app/infra developer and the project manager. We split the general group of citizen scientists into user/contributor and the volunteer expert. Although both fall under the CS umbrella, they may perform different tasks because their skill and engagement levels differ. The citizen- and professional researcher, computer scientist and app/infra developer each are domain experts, and their roles largely correspond to the research, machine learning development and app/infra development subject domains respectively. However, their tasks will sometimes overlap as input or support from different domains may be required. The project manager is involved with all subject domains in the project and is responsible for keeping track of general progress and the different objectives and challenges. Here, we also group the previously mentioned liaison officer role with the project manager role.

App and infrastructure

An online CS platform requires a durable and adaptable infrastructure to reach a CS-ML project’s full scientific and public engagement potential. Specific app and infrastructure designs depend on the wants and needs of particular CS-ML project participants. Here, we summarise findings from the reviewed articles in a generalised app and infrastructure workflow and relate the different aspects of the workflow to the previously identified CS-ML approach subject domains (Fig. 8). Our generalised workflow consists of a mobile/web app, set of web services for deploying ML models and storing data, a database web portal and a linked open data service.

The mobile/web app is the main front end of the workflow and is designed for engagement with citizen scientists and other stakeholders. The app aims at a user-friendly environment with clear instructions for consistent high-quality data. It may contain information on the project and participation guidelines as well as communication and feedback functions to interact with other contributors and the project managers (e.g., chat, forum, mail, newsletters, messages). Furthermore, it contains the ML tools for e.g., image classification, and a portal for volunteer experts to validate the data and ML model predictions. Finally, the app contains different tools to engage with collections and data that is gathered through the app. These tools can include a personal collection registration system, a digital reference collection, data overviews such as graphs and maps and additional subject information. The app is where contributors from all CS-ML approach subject domains interact and engage with the project.

The set of web services are the backbone of the infrastructure where data and ML models are stored and through which the data flows between different front-end portals. Depending on the individual project needs, the web services can include APIs for storing new observations, (reference) database and ML model servers. For the web service infrastructure design, input is required from different approach domains such as research and ML development. As designing and building an infrastructure from scratch is expensive and time consuming, it might be worth considering making use of existing, large infrastructures. An additional benefit from a shared, large infrastructure is that there might be pre-existing audiences that can be introduced to your project.

The database web portal and linked open data service do function as research tools with which the data can be analysed, compared to other datasets and used to develop new tools. In the database web portal, citizen scientists, researchers and other stakeholders can go through, filter and visualise data from the platform database. These tools can be used to gain general insights, produce research or create educational material. The linked open data service is mainly geared towards researchers, data scientists and software developers that want to combine the data from different databases to create further applications or conduct further data research.

Specific challenges in object-based CS-ML approaches: lessons from the LegaSea project

Finally, we discuss aspects of the implementation of our proposed CS-ML workflow in practice with a case study, the NWO-ENW-funded LegaSea project. LegaSea aims to identify and characterise late-Quaternary vertebrate communities from rich beach finds through a CS-ML approach and is currently in the execution phase. Late-Quaternary fossils and artefacts are found in strata underlying the seafloor in the southern North Sea, the area known as ‘Doggerland’ (Amkreutz & Van der Vaart-Verschoof, 2021). These strata provide sand used for beach nourishments in the Netherlands that include the abundant fossils and artefacts. The publicly accessible sand nourished beaches attract lots of CS activity, outreach and scientific output (e.g., Amkreutz & Van der Vaart-Verschoof, 2021; Kuitems et al., 2015; Moerdijk et al., 2010; Mol, 2016; Mol et al., 2006; Mol & Bakker, 2022). Active collaboration between citizen scientists, researchers and institutes is organised through societies and working groups (e.g., Working Group Pleistocene Mammals, WPZ, and Working Group Tertiary and Quaternary Geology, WTKG) as well as through online platforms (e.g., Oervondstchecker; https://www.oervondstchecker.nl/) and social media groups (e.g., ‘Strandfossielen’ on Facebook). The Dutch CS community is very diverse and ranges from participants that operate on a professional level to novice collectors with little experience. Engagement must be tailored to specific subgroups of citizen scientists. Furthermore, CS (field) days are organised regularly by societies, institutes and museums such as the Dutch National Museum of Antiquities and Futureland.

In the preparation phase of the LegaSea project, citizen scientists, researchers and computer scientists that are engaged with the research subject and/or host institute were consulted on the project aims and outline. This way the main objectives and challenges for the CS tasks, research and ML development were addressed. There are several CS communities, networks and initiatives working on Quaternary beach finds with active collaborations with professional researchers, but these communities lack an overarching platform with a robust and open data framework which would benefit the collaboration between researchers and citizen scientists. Within the Netherlands fossil finds are not regulated. However, archaeological artefacts are subject of a multitude of heritage laws such as formulated in the Valletta Convention (Council of Europe, 1992) and the Dutch Heritage Act (Government of the Netherlands, 2016) that aim to preserve archaeological heritage and only allow excavation under specific circumstances and under strict conditions (Van Kolfschoten, 2006). For the beach finds that are ex-situ and do not derive from land, most of these laws are not applicable other than the requirement to administer archaeological finds. Only for human remains (including fossil human remains) do strict regulations exist for reporting material to the authorities. Discouraging beach collecting through overregulation would be counterproductive as these objects will disappear when not collected. However, application of our CS-ML workflow to in-situ on land sites will need further steps to incorporate legal requirements for such activities. This is beyond the scope of LegaSea. A pre-existing ML infrastructure at the host institute, designed and used for biodiversity monitoring, could be used and adapted in the LegaSea project. This ‘pipeline’ allowed for a quick exploration and piloting phase which informed the data collection strategy in the execution phase of the project.

Both the stakeholder engagement and the app and infrastructure development were not yet outlined in the preparation phase of the project. First, the specific needs and requirements form the target CS communities had to be explored and established. Within the LegaSea project we organized a CS participant board and discussed with them their uses, requirements and desires for ML-based support and an online platform. Another stakeholder research relevant for the project is currently being carried out under the umbrella of the NWO-SWG funded research program ‘Resurfacing Doggerland’, which targets the post-glacial human occupation history of the southern North Sea, engages with CS and uses in part beach finds of both artefacts and fossils. Citizen scientists and other stakeholders were asked about their previous experiences working with scientists and online CS platforms as well as their wants and needs for a future online CS platform. The findings are presented and discussed in the ‘Doggerland Heritage Community’ (DOGHER-C) report (S van der Vaart-Verschoof, 2025, in preparation) and provide valuable insights in the different experiences of the citizen scientists and other stakeholders. A common approach is desirable as these objects derive from the same context and provide relevant and complementary information for both archaeology and palaeontology (Fig. 1).

So far, the main emphasis of the execution phase of LegaSea has been on data collection, validation and pre-processing. To increase public engagement, part of the dataset was photographed at a public venue at the host institute over a period of 2 months. Further complementary datasets were collected from other institutes in the Netherlands, as well as from CS collections. With the help of citizen scientists and a volunteer at the host institute, there is a continuous generation of new data. As both data quality and quantity affect the model performance, the quality of data from different sources is currently being investigated as part of the workflow by comparing model performances on photographs taken under ideal conditions and photographs uploaded to an online CS platform (https://www.oervondstchecker.nl/). Furthermore, we are currently assessing whether various data augmentation techniques can improve model performance. A major difference between observation-based biodiversity and object-based archaeological and palaeontological CS-ML approaches is the nature of metadata. Important data in the former are location and time of observation and furthermore context data such as habitat or environmental parameters. Important additional data in the latter are data of the stratigraphic source (geological unit, age, interpreted palaeoenvironment). In the case of ex-situ beach finds these require geological research in the extraction areas (e.g., Niekus et al., 2023) and often come with uncertainties such as diffuse age-data (e.g., Busschers et al., 2014). Providing such context data often is beyond the remit of individual CS (but there are exceptions, e.g., Mol, 2016) and requires professional support or management.

The LegaSea project has not yet reached the implementation and reiteration phases, but a start has been made with publishing the ML models from the project. Doing so early in the project allows us to motivate and galvanise citizen scientists and stakeholders to participate in the project and to provide further (data) input and feedback consistent with an Open Science approach. This early implementation benefits from the pre-existing infrastructure at the host institute. Further implementation and continuation of the project results require additional, structural financial support and institutional embedding. Exploration of the possibilities requires an approach where all stakeholders are involved and feel like the project is mutually beneficial. Our CS-ML approach addresses those needs and makes a successful project possible.

Conclusions

Object-based research in paleontological and archaeological domains hugely benefits or is even entirely dependent on the contribution of collectors and citizen scientists. The synergy between CS and ML creates many opportunities for the collection/generation, validation and analysis of large datasets and opens new avenues of public engagement and participation. This new interdisciplinary field currently lacks a streamlined approach, and projects might therefore not achieve their full scientific and public outreach potential. In this review, we formulate a workflow for research projects with a CS-ML approach based on best practices in fields outside palaeontology and archaeology and common pitfalls from the literature and the authors’ experiences. The approach consists of four phases: (I) preparation, (II) execution, (III) implementation and (IV) reiteration. We recognise five subject domains (CS tasks, ML development, research, stakeholder engagement and app/infrastructure development) which each have their own objectives and challenges. Because a clear subdivision of tasks is essential to structure and manage the workflow, tasks are allocated to project contributors in different roles. Moreover, an outline for an integrated online CS platform will help reach the projects full scientific and public outreach potential. In practice, CS-ML projects might differ substantially in the preparation phase as specific preconditions dictate which objectives and challenges require most attention. Project managers have to identify those needs through stakeholder engagement to make a successful project possible. Our proposed workflow provides additional elements to transfer successful observation-based approaches common in the biodiversity domain to more object-based approaches in the palaeontological-archaeological domains.

Supplemental Information

Supplemental Information 1 General characterisation of reviewed articles.

Supplemental Information 2 Results, workflows and suggestions for workflows and further research from reviewed articles.

This research has benefited from discussions with and help from the project supervision board, including Bram Langeveld, Hansjorg Ahrens and Natasja den Ouden. Furthermore, we want to thank Vincent Kalkman, whose insights improved the manuscript. We thank the anonymous reviewers for their very concise and helpful suggestions. Finally, we want to thank the many Dutch palaeontological and archaeological collectors and citizen scientists involved in the project.

Additional Information and Declarations

Competing Interests

The authors declare that they have no competing interests.

Author Contributions

Isaak Eijkelboom conceived and designed the experiments, performed the experiments, analyzed the data, prepared figures and/or tables, authored or reviewed drafts of the article, and approved the final draft.

Anne S. Schulp conceived and designed the experiments, analyzed the data, authored or reviewed drafts of the article, and approved the final draft.

Luc Amkreutz analyzed the data, authored or reviewed drafts of the article, and approved the final draft.

Dylan Verheul analyzed the data, authored or reviewed drafts of the article, and approved the final draft.

Wouter Verschoof-van der Vaart analyzed the data, authored or reviewed drafts of the article, and approved the final draft.

Sasja van der Vaart-Verschoof analyzed the data, authored or reviewed drafts of the article, and approved the final draft.

Laurens Hogeweg analyzed the data, authored or reviewed drafts of the article, and approved the final draft.

Django Brunink analyzed the data, prepared figures and/or tables, authored or reviewed drafts of the article, and approved the final draft.

Dick Mol analyzed the data, authored or reviewed drafts of the article, and approved the final draft.

Hans Peeters analyzed the data, authored or reviewed drafts of the article, and approved the final draft.

Frank Wesselingh conceived and designed the experiments, performed the experiments, analyzed the data, prepared figures and/or tables, authored or reviewed drafts of the article, and approved the final draft.

Data Availability

The following information was supplied regarding data availability:

This is a literature review.

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
