# Peer review of "Making sense of fossils and artefacts: a review of best practices for the design of a successful workflow for machine learning-assisted citizen science projects"

_PeerJ, doi:10.7717/peerj.18927_

## Round 0.1 · original submission · Major Revisions

Please, address all the reviewers' comments and questions. Notice that they raised concerns regarding the study design and the choice of literature.

Reviewer 1 ·

Basic reporting

The manuscript meets all basic reporting criteria except for some references/background that should be added (see Additional Comments for full details).

Experimental design

In my view, there are two issues with the study design: (1) The lack of depth in the analysis (particularly the lack of numbers throughout) and (2) the lack of specificity of much of the discussion towards palaeontology and archaeology, which is the stated purpose of the article. I have discussed these issues in detail in the Additional Comments section.

Validity of the findings

I believe the manuscript meets all of these standards.

Additional comments

In their manuscript, "Making sense of fossils and artefacts: A review of best practices for the design of a successful workflow for machine-learning assisted citizen science projects", Eijkelboom et al. review the state of the emerging field of citizen science combined with machine learning (termed the CS-ML approach), specifically within the context of the use of this approach in the fields of palaeontology and archaeology. The subject of the manuscript is very timely and the manuscript is very well-written and clear, summarising the breadth of work in this field. The recommendations about best practices are concrete and useful, and will be a helpful guide for researchers interested in the CS-ML approach. However, there are a couple of major issues that I believe need to be addressed before acceptance and publication, namely:

1. The review of previous work has good breadth, but is quite shallow. In particular, there is a notable and pervasive lack of numbers and explicit comparisons that make it very difficult to evaluate the state of the field and the efficacy of techniques, particularly in comparison to the closely related field of biodiversity monitoring at large. This is a major omission since this paper focuses specifically on the CS-ML approach as applied to palaeontology and archaeology (which is what distinguishes it from previous work on CS-ML and biodiversity monitoring).

For example, on lines 238-241, a comparison of error rates in expert- vs. CS-generated datasets is relevant and an important omission in this discussion, particularly since the authors repeatedly state that mistrust in CS datasets is a major roadblock in their usage. In general, the paper would greatly benefit from more numbers when discussing improved performance from various strategies (e.g., lines 283-303). How big are these improvements? What are the baselines? In line 424, the authors state that "clear instructions and rewards can improve participation and data quality" -- by how much? Increasing from 10 to 11 participants, for example, is technically improved participation but not meaningfully so. The lack of numbers throughout makes these qualitative statements difficult to evaluate, which in turn makes it difficult to meaningfully assess the impact of the listed strategies for improvement. (As a side note, I think a table of some sort summarizing all the strategies discussed would be very useful.) Very basic information along these lines is missing throughout, even from the LegaSea case study (such as how many citizen scientists were involved).

2. Overall, the suggestions the authors make for best practices, while thorough and useful, are not really specific to palaeontology and archaeology. Rather, they are general recommendations that are applicable for all CS-ML studies, particularly biodiversity monitoring initiatives. (In fact, the LegaSea case study itself uses a repurposed biodiversity monitoring ML pipeline. Reusing infrastructure when possible is obviously good practice; I only bring this up to illustrate lack of distinction between the purported focus of this paper -- palaeontology and archaeology -- and biodiversity monitoring.) This is not a problem in and of itself, per se, particularly since PeerJ is not concerned with an assessment of novelty. However, it does make the core framing of the paper -- this it is focused on palaeo/archaeo CS-ML approaches specifically -- a bit misleading. More comparisons between palaeo-/archaeo-specific considerations and biodiversity initiatives as a whole are necessary to justify the current framing of the paper. For instance, are there differences between motivating the public to engage with palaeo-/archaeo-focused projects, vs. projects that focus on extant biodiversity? Numbers with regards to number of participants in palaeo/archaeo CS-ML initiatives vs. extant biodiversity initiatives would help a lot in this regard (i.e., addressing the problem I outlined in the first point, above). The authors do mention that local laws are relevant (such as distinctions between the treatment of palaeontological vs. archaeological samples), but this is simply mentioned without analysis (e.g., which countries do more palaeo vs. archaeo CS-ML work? Does this correlate with regulations/laws?).

The following are specific points I noticed, many of which are encompassed within the two big points above. (Apologies if there is some repetition, as these are notes I made while reading and before I wrote the summaries above.)

- Lines 224-229: I think this discussion should touch on the process of acquiring enough volunteers in the first place. Relatedly, information on the number of CS volunteers for each study would be quite interesting, perhaps correlated to how which techniques were used to encourage participating. Overall, this paper has only very qualitative discussions of the field, which diminishes its impact and usefulness.

- Do your search strings catch studies that are conducted using services such as Mechanical Turk, or do you consider paid services to be outside the scope of this study? Or is there no distinction made? If they are included, does paying vs. not paying have an effect on performance (as suggested in previous studies, e.g., Klie et al. 2023, Gandhi et al. 2023)?

- Line 247-249: This has been identified in earlier work e.g. Hsiang et al. 2019, although in the context of expert identification rather than CS. This bias was significantly stronger for humans compared to the machine. In general, I think there should be more discussion/citation of literature related to labelling accuracy, e.g., Austen et al. 2016, Kallimanis et al. 2017, Bennett et al. 2023), particularly in relation to comment #1 -- that is, if we don't know how well experts perform at these sorts of tasks, we have no baseline by which to evaluate how well citizen science volunteers perform at these tasks.

- In the infrastructure section (starting line 435), I think it be useful to list the infrastructures used and how often they are used. For example, what percentage of the reviewed papers use, say, Zooniverse? This would help answer questions such as, how many studies build platforms from scratch (a potential waste of time/resources?), which platforms seem to be popular and work well, etc. This information is probably best conveyed in a table. Additional information from the text could also be tracked in this table (e.g., which platforms allow for offline data collection, etc.)

- The font size in figures 6 and 7 is too small. The white text on the red/blue/green bubbles is also difficult to read. I'd suggest changing the colors so that all text is black, or adding a stroke to the white text to make it thicker and easier to read. There are also typos in some of the bubbles (e.g., second blue bubble under Implementation/Reiteration in the Project Manager column says 'Instituational' instead of 'Institutional').

- It may be worthwhile to mention/discuss large biodiversity databases that have citizen science contributions, such as GBIF, in the introductory sections.


Citations:
Klie, Jan-Christoph et al. “Lessons Learned from a Citizen Science Project for Natural Language Processing.” Conference of the European Chapter of the Association for Computational Linguistics (2023).

Gandhi, K., Spatharioti, S. E., Eustis, S., Wylie, S., & Cooper, S. (2022). Performance of Paid and Volunteer Image Labeling in Citizen Science — A Retrospective Analysis. Proceedings of the AAAI Conference on Human Computation and Crowdsourcing, 10(1), 64-73. https://doi.org/10.1609/hcomp.v10i1.21988

Hsiang, A.Y., Brombacher, A., Rillo, M.C., Mleneck-Vautravers, M.J., Conn, S., Lordsmith, S., Jentzen, A., Henehan, M.J., Metcalfe, B., Fenton, I.S., Wade, B.S., Fox, L., Meilland, J., Davis, C.V., Baranowski, U., Groeneveld, J., Edgar, K.M., Movellan, A., Aze, T., Dowsett, H.J., Miller, C.G., Rios, N. and Hull, P.M. (2019), Endless Forams: >34,000 Modern Planktonic Foraminiferal Images for Taxonomic Training and Automated Species Recognition Using Convolutional Neural Networks. Paleoceanography and Paleoclimatology, 34: 1157-1177. https://doi.org/10.1029/2019PA003612

Austen, G., Bindemann, M., Griffiths, R. et al. Species identification by experts and non-experts: comparing images from field guides. Sci Rep 6, 33634 (2016). https://doi.org/10.1038/srep33634

Kallimanis, A.S., Panitsa, M. & Dimopoulos, P. Quality of non-expert citizen science data collected for habitat type conservation status assessment in Natura 2000 protected areas. Sci Rep 7, 8873 (2017). https://doi.org/10.1038/s41598-017-09316-9

Bennett, A.F., Haslem, A., White, M., Hollings, T., & Thomson, J.R. (2023). How expert are 'experts'? Comparing expert predictions and empirical data on the use of farmland restoration sites by birds. Biological Conservation, 282: 110018. https://doi.org/10.1016/j.biocon.2023.110018



Minor Comments:

Line 53: 'Neanderthal' not 'Neandertal'

Line 80: 'lowers' not 'lower'

Line 118-119: Change to 'The latter have become more professional over time in both collecting and documenting, becoming a major source of scientific data and materials.'

Line 120: 'volume' not 'volumes'

Line 121: Change to 'the past 150 years (and which continue to be found) form a suitable dataset for ML'

Line 124: 'communicate' not 'communicates'

Line 124: Comma after 'importantly'

Line 147: I think it should be 'in the included search strings' if I understand this sentence correctly?

Figure 3 (and throughout manuscript): I believe numbers in PeerJ are formatted with commas as the thousands separator rather than periods.

Figure 4: Please add x- and y-axis labels, and a legend (i.e., make the figure fully understandable on its own and at at glance).

Line 187: 'spread thinly' not 'thin spread'

Line 254: 'assess' not 'asses'

Line 312: 'originate' not 'originates'

Reviewer 2 ·

Basic reporting

The manuscript entitled “Making sense of fossils and artefacts: A review of best practices for the design of a successful workflow for machine learning-assisted citizen science projects” is a really interesting contribution on the knowledge of the usefulness of machine learning citizen scientist approaches. A literature review article is always a challenge, to be comprehensive and to analyse all factors. This work is a good overview of the state of the art and provides a useful guide for developing this methodological approach. However, as most of the valuable manuscripts, this study also needs some revisions and improvements to be published.

I believe that legal issues are the key problematic factors in citizen science approaches as the one proposed here. Not only differences in laws between fossils and archaeological artefacts, but also with regard to the protection of each of them. I think that the approach of analyzing isolated artefacts or fossils by citizens involves relevant risks for their protection, since to preserve stratigraphic and archaeological context is essential for the development of appropriate studies. This will hardly be preserved if the recovery or remains is not carried out under the supervision of specialized researchers, following the appropriate record methodologies. I think that it is very much needed to take this issue in consideration in the manuscript.

Line 67
An important application of the use of ML in Palaeontology is taxonomic classification, therefore it should be mentioned and some references added (eg. Wills et al. 2021; Moclán et al. 2023)
Wills, S., Underwood, C.J., Barrett, P.M., 2021. Learning to see the wood for the trees: machine learning, decision trees, and the classification of isolated theropod teeth. Palaeontology 64, 75-99. https://doi.org/10.1111/pala.12512.
Moclán A, Domínguez-García ÁC, Stoetzel E et al (2023) Machine learning interspecific identification of mouse first lower molars (Genus Mus Linnaeus, 1758) and application to fossil remains from the Estrecho Cave (Spain). Quat Sci Rev 299:107877. https://doi.org/10.1016/j.quascirev.2022.107877

Lines 138-140

“palaeontology and archaeology were grouped together, as there is overlap in (the nature of) studied materials and methods”

In my opinion, this sentence needs to be reviewed and a deeper explanation should be done. Although palaeontology and archaeology share common methodologies and the nature of material in some cases (vertebrate bones), there is major differences to consider. It is true that in Quaternary fossil sites showing human evidences, researchers from both disciplines work together in multidisciplinary teams. However, palaeontological studies cover much more chronologies and materials (eg. invertebrates, palaeobotany), as well as archaeology studies remains unrelated to palaeontology (lithic tools, pottery). Thus, this should be reviewed and clarified, as a better delimitation of artefacts and fossils considered is necessary.

Lines 172-174

I do not understand why Table S1 contains 69 articles and table S2 contains 80 articles.

Line 760

Could you provide the reference of the ML models publication?

Experimental design

It is a bit confusing the fact of including many reviewed articles that do not address fossils and/or artefacts. I understand that they were included as references on ML and SC applications. Maybe a reorganization of the database of the reviewed articles, including different categories (e.g. relevance categories of table 2) would be useful. This will therefore have to be explained in the methods section.

In addition, from my area of expertise I detect certain articles included in this work do not fit well with any inclusion category according to their relevance. For example, articles on ML projects using data which is unlikely attainable by CS (eg. Courtenay and González Aguilera, 2020; Moclán et al. 2020; Vegara-Riquelme et al. 2023), such as Geometric Morphometric data specialized notches classification or using microscope magnification. There is a wide range of excellent work in this line of research, which demonstrate the great usefulness of ML in archaeological studies and should be mentioned in this paper. However, these approaches are not applicable to SC data. In that case, they should be limited to ML studies that work directly using images (not obtained using stereomicroscope).

Concerning the organization of manuscript in different sections, I suggest to move the phased workflow designed to results instead of discussion section. On the contrary, the subsection 3.2 “The CS-ML approach: potential and pitfalls” seems to me more appropriate to consider as discussion than results. However, I think that this section should be improved highlighting methods and processes involved specifically in the CS-ML approach applied to paleontology and archeology.

Validity of the findings

no comment

---

## Round 0.2 · Minor Revisions

Thank you for incorporating the previous reviewers' suggestions and answering the comments. The manuscript will be ready for publication after some final minor corrections pointed out by one of the reviewers.

Reviewer 1 ·

Basic reporting

The authors have addressed my previous concerns regarding reporting.

Experimental design

The authors have addressed my previous concerns regarding study design. The paper is much improved by the additional of quantitative information and tables 4-6.

Validity of the findings

No comment.

Additional comments

I include here some minor language comments and also a suggestion for the presentation of tables 4-6.

Line 41 - 'than' not 'then'

Line 149 - 'practiced' not 'practice'

Line 299-301 - Reword to: 'These general pros, cons and "lessons learned" for the CS-ML approach, which have emerged over recent years from fields such as biodiversity monitoring, inform our CS-ML approach in palaeontology and archaeology'

Line 377 - Remove extra space after (Mugford et al., 2021)

Line 381 - 'an' not 'a'

Line 390 - 'yields' not 'yield'

Line 390-391 - Change 'more consistently high' to 'consistently higher'

Line 896 - remove comma after 'beach finds'

Line 898 - Add 'do' before 'strict regulations'

Line 902 - Make 'this is beyond the scope of LegaSea' its own sentence

Line 941-942 - Reword 'often come with uncertainties as for example age-data are diffuse' to 'often come with uncertainties such as diffuse age-data'

Line 949 - I'm not sure 'lead' is the correct word here -- maybe 'leads' or 'leadership'?

Line 968 - Remove extra space after 'practices'

Tables 4, 5, 6 - I wonder if chronological order for the studies might be better than alphabetical, as then it gives the reader a sense of how the methods are evolving/building over time (rather than just an arbitrary order).

Reviewer 2 ·

Basic reporting

In the revised version of the manuscript, the authors have greatly improved the text and incorporated the reviewer's suggestions appropriately.
Therefore, I have no further comments and would like to congratulate the authors on the well work done, proposing the acceptance of the article.

Experimental design

no comment

Validity of the findings

no comment

---

## Round 0.3 · accepted · Accept

Thank you for addressing all reviewer's comments. I am happy with the current version and in my opinion the manuscript is ready for publication.